# On the Hardness of Computing Counterfactual and Semi-factual Explanations in XAI

**André Artelt**                                                     *aartelt@techfak.uni-bielefeld.de*
*Faculty of Technology*
*Bielefeld University, Germany*

**Martin Olsen**                                                               *martino@btech.au.dk*
*Department of Business Development and Technology*
*Aarhus University, Denmark*

**Kevin Tierney**                                                       *kevin.tierney@univie.ac.at*
*Department of Business Decisions and Analytics*
*University of Vienna, Austria*

**Reviewed on OpenReview:** *https://openreview.net/forum?id=aELzBwOq1O*

## Abstract

Providing clear explanations to the choices of machine learning models is essential for these models to be deployed in crucial applications. Counterfactual and semi-factual explanations have emerged as two mechanisms for providing users with insights into the outputs of their models. We provide an overview of the computational complexity results in the literature for generating these explanations, finding that in many cases, generating explanations is computationally hard. We strengthen the argument for this considerably by further contributing our own inapproximability results showing that not only are explanations often hard to generate, but under certain assumptions, they are also hard to approximate. We discuss the implications of these complexity results for the XAI community and for policymakers seeking to regulate explanations in AI.

## 1 Introduction

As machine learning (ML) models are increasingly used in critical applications, the ability to provide explanations about their outputs becomes crucially important. The EU AI Act (European Parliament and Council of the European Union, 2024) adds a regulatory motivation to enabling the explanation of models, as it "gives those subject to an algorithmic decision the right to an explanation of that decision" (Nisevic et al., 2024). The field of explainable artificial intelligence (XAI) has emerged to address these challenges (Samek & Müller, 2019; Minh et al., 2021).

However, despite the focus on providing explanations both in the research community and regulatory bodies, the computational complexity of explanations has only just begun to be explored. This raises the central question of this work, namely, under what conditions can we obtain explanations and in what computational complexity? The answer to this question has an impact beyond the research domain, potentially affecting the feasibility of future laws and regulation concerning explainability in AI (Verma et al., 2024a).

One key focus of the XAI community is on *counterfactual* explanations (Guidotti, 2024), which describe how changing the input features to an ML model changes the output prediction of this model. An alternative to counterfactual explanations are *semi-factual* explanations (Aryal, 2024), which highlight changes in input features that *do not* lead to changes in the model output. Both of these types of explanations allow users of models to understand how changing input affects their model, providing increased transparency, trust and potentially better decision making.

The computational complexity of generating counterfactual and semi-factual explanations varies greatly depending on the ML model and type of explanation desired, and in many cases is unknown. The literature has recently made great strides in defining various cases of explanations and providing proofs of computational complexity. However, the literature is split between different communities, using different terminology and different formalizations, making it challenging to get a comprehensive overview. In particular, there is no unifying framework to unify the many directions of research to give a full view of the current state of the complexity of explanations.

This paper aims to resolve this and provides the following contribution.

1. We provide a comprehensive and accessible overview of the literature analyzing the computational complexity of counterfactual and semi-factual explanations.

2. We study the existence of approximation algorithms in several explanation settings, further refining the existing knowledge of the complexity of explanations.

3. We provide a critical discussion of our findings and what they mean for the XAI community.

This paper is organized as follows. We first provide some background about counterfactual explanations and foundations in computational complexity in Section 2. Next, we provide a framework of the complexity of explanations for various types of ML models, detailing both what we currently know and open challenges in Section 3. In Section 4 we provide novel approximation results for some explanation settings, whereby all proofs can be found in Appendix A.2. Finally, in Section 5 we discuss open questions and implications of computational complexity analysis for the XAI community.

## 2 Background & Foundations

### 2.1 Counterfactual & Semi-factual Explanations

A counterfactual explanation (CFE), often referred to simply as a *counterfactual*, outlines the specific changes that can be made to the features of a particular instance to alter the system's output. Typically, such explanations are sought when the outcome is unexpected or unfavorable (Riveiro & Thill, 2022). In the latter case, a counterfactual is also referred to as *(computational) recourse* (Karimi et al., 2023), i.e. recommendations on how to change the unfavorable into a favorable outcome. Because counterfactuals can mimic ways in which humans explain (Byrne, 2019), they constitute one of the most popular explanation methods in literature and in practice (Molnar, 2019; Verma et al., 2024a).

Counterfactual explanations (Wachter et al., 2017) have two key characteristics: 1) the contrasting property, which necessitates a change in the system's output, and 2) the cost of the counterfactual, meaning the effort and resources required to implement the counterfactual in the real world should be minimized to enhance its practicality. This often involves making as few changes as possible or ensuring that the changes are minimal. Both properties can be combined into an optimization problem (see Definition 1).

**Definition 1** ((Classic) Counterfactual Explanation)**.** *Assume a classifier (binary or multi-class) $h : \mathcal{X} \to \mathcal{Y}$ is given. Computing a counterfactual $\delta_{cf}$ for a given instance $\boldsymbol{x}_{orig} \in \mathcal{X}$ is phrased as the following optimization problem:*

$$\underset{\delta_{cf}}{\arg\min} \ \theta(\delta_{cf}) \quad s.t. \ h(\boldsymbol{x}_{orig} \oplus \delta_{cf}) = y_{cf} \tag{1}$$

*where $\theta(\cdot)$ denotes the cost of the explanation (e.g., cost of recourse) that should be minimized. We refer to the final counterfactual sample $\boldsymbol{x}_{orig} \oplus \delta_{cf}$ as $\boldsymbol{x}_{cf}$.*

**Remark 1.** *An alternative to the constrained optimization problem Eq. 1 is to use a single objective weighting the contrastive and cost properties as proposed by Wachter et al. (2017):*

$$\underset{\delta_{cf}}{\arg\min} \ \ell(h(\boldsymbol{x}_{orig} \oplus \delta_{cf}), y_{cf}) + \lambda \cdot \theta(\delta_{cf}) \tag{2}$$

*where $\lambda > 0$ denotes a regularization parameter that balances the two objectives, and $\ell(\cdot)$ denotes a suitable loss function such as the zero-one loss or the squared error in the case that $h(\cdot)$ denotes a regressor instead of a classifier.*

To not make any assumptions on the data domain, we use the symbol $\oplus$ to denote the application/execution of the counterfactual $\delta_{\mathrm{cf}}$ to the original instance $\boldsymbol{x}_{\mathrm{orig}}$. While in the case of real and integer numbers (e.g., $\mathcal{X} = \mathbb{R}^d$) this reduces to a translation, i.e., $(\boldsymbol{x}_{\mathrm{cf}})_i = (\boldsymbol{x}_{\mathrm{orig}})_i + (\delta_{\mathrm{cf}})_i$, in the case of categorical features it denotes a substitution, i.e. $(\boldsymbol{x}_{\mathrm{cf}})_i = (\delta_{\mathrm{cf}})_i$.

Note that the cost of the counterfactual (often also referred to as the *cost of recourse*), here modeled by $\theta(\cdot)$, is highly domain and use-case-specific. It therefore must be chosen carefully in practice, and might require domain knowledge. In many implementations and toolboxes (Guidotti, 2024), the *p*-norm is used as a default.

Besides those two essential properties (contrasting and cost), additional relevant aspects exist such as plausibility (Artelt & Hammer, 2020; Van Looveren & Klaise, 2021; Poyiadzi et al., 2020), diversity (Mothilal et al., 2020), robustness (w.r.t. to input perturbations or model changes) (Artelt et al., 2021; Zhang et al., 2023; Leofante & Potyka, 2024; Marzari et al., 2024), and fairness (Artelt & Hammer, 2023; von Kügelgen et al., 2022; Sharma et al., 2021; 2020), which have been addressed in literature (Guidotti, 2024). In particular, robustness with respect to model changes has recently received significant attention (Marzari et al., 2024; Leofante & Wicker, 2025). Here, a set $\triangle\mathcal{H}(h)$ is assumed that contains all plausible variations/changes of a given classifier $h(\cdot)$. We can then extend Eq. 1 for computing a counterfactual $\delta_{\mathrm{cf}}$ that is robust under those model changes $\triangle\mathcal{H}(h)$ as follows (Marzari et al., 2024):

$$\underset{\delta_{\mathrm{cf}}}{\arg\min}\ \theta(\delta_{\mathrm{cf}}) \quad \text{s.t.}\ h'(\boldsymbol{x}_{\mathrm{orig}} \oplus \delta_{\mathrm{cf}}) = y_{\mathrm{cf}} \quad \forall h' \in \triangle\mathcal{H}(h) \tag{3}$$

However, it is worth noting that the basic formalization in Eq. 1 is still very popular and widely used in practice (Verma et al., 2024a; Guidotti, 2024).

There exist numerous methods (mostly heuristics) (Guidotti, 2024) for computing counterfactual explanations, i.e., computing feasible solutions to Eq. 1. Gradient-based methods for solving Eq. 2 (Wachter et al., 2017), assuming a differentiable model, are extremely popular in the literature (Guidotti, 2024). For non-differentiable models, such as tree-based models, evolutionary methods (Mothilal et al., 2020; Dandl et al., 2020) are commonly used. Furthermore, evolutionary methods also constitute the most popular choice in the case of categorical features, which often occur in real-world scenarios such as attrition or business analytics (Artelt & Gregoriades, 2023; 2024). However, those existing methods usually compute a feasible solution only and are unable to guarantee optimality. In this context, a study of the computational complexity of the corresponding optimization problems being solved can provide insight into the cases in which it is (or is not) possible to (efficiently) compute optimal or approximately optimal counterfactual explanations.

It is important to note that Definition 1 represents a non-causal approach, meaning it does not involve modeling underlying causal mechanisms. A separate branch of research on counterfactuals exists that uses structural causal models to integrate causal knowledge (Karimi et al., 2020). However, in practice, such causal models are often unknown and must be estimated from data or precisely defined with the assistance of domain experts. This work focuses solely on this non-causal approach because all of the literature on the computational complexity of counterfactual explanations currently only considers the non-causal case.

As an alternative to Definition 1, counterfactual explanations are also related to the *Minimum Change Required (MCR)* considered in the computational complexity literature (Barceló et al., 2020). Here, the existence of a counterfactual $\delta_{\mathrm{cf}}$ with an upper bound $k$ on its cost $\theta(\delta_{\mathrm{cf}})$ is stated as a *decision problem*:

$$\text{Does there exist a } \delta_{\mathrm{cf}} \text{ s.t. } h(\boldsymbol{x}_{\mathrm{orig}} \oplus \delta_{\mathrm{cf}}) = y_{\mathrm{cf}} \text{ and } \theta(\delta_{\mathrm{cf}}) \leq k? \tag{4}$$

Note that if the decision problem Eq. 4 is computationally hard, so is the corresponding optimization problem Eq. 1 (Barceló et al., 2020).

In the context of plausible counterfactual explanations, Amir et al. (2024) assumes the availability of an additional function (called *context indicator*) $\pi : \boldsymbol{x}_{\mathrm{orig}} \oplus \delta_{\mathrm{cf}} \mapsto \{0, 1\}$ for distinguishing between plausible

($\pi(\cdot) = 1$) and non-plausible ($\pi(\cdot) = 0$) counterfactuals. This context indicator $\pi(\cdot)$ is the added as an additional constraint to Eq. 4 to ensure plausible counterfactual explanations:

$$\text{Does there exist a } \delta_{\text{cf}} \text{ s.t. } h(\boldsymbol{x}_{\text{orig}} \oplus \delta_{\text{cf}}) = y_{\text{cf}} \text{ and } \pi(\boldsymbol{x}_{\text{orig}} \oplus \delta_{\text{cf}}) = 1 \text{ and } \theta(\delta_{\text{cf}}) \leq k? \tag{5}$$

However, it is worth noting that apart from the plausibility formulation Eq. 5 studied in (Amir et al., 2024), there exist numerous other plausibility formulations, such as causal plausibility (Karimi et al., 2023) that have not been considered so far when analyzing the computational complexity of computing counterfactual explanations.

### 2.1.1 Semi-factual Explanations

Semi-factual explanations (often just called *semi-factuals*) (Aryal & Keane, 2023; Aryal, 2024; Kenny & Huang, 2023), also known as "even if" explanations, highlight input changes that, unlike counterfactuals, *do not alter the outcome* – i.e., "Even if X and Y had been different, the outcome would remain unchanged". These explanations provide a rationale for counterfactuals by illustrating which changes would not affect the outcome, helping users understand what modifications would leave the outcome intact. Compared to counterfactuals, semi-factuals have received much less attention in the XAI community (Aryal & Keane, 2023). The existing research on semi-factuals in XAI explores the explanation of reject options (Artelt & Hammer, 2022), computational approaches for plausible semi-factual and counterfactuals (Kenny & Keane, 2021), as well as connecting semi-factuals and counterfactuals on a computational level (Aryal & Keane, 2024).

Unlike counterfactual explanations, there is no universally accepted formalization for semi-factual explanations. Formalizing semi-factual explanations requires balancing two conflicting objectives (Aryal & Keane, 2024; 2023): 1) Maximizing the changes made to the inputs without altering the outcome, and 2) Minimizing the number of changes to the inputs to facilitate easier human understanding. While some approaches look for sparse changes such that the distance to the decision boundary stays the same or becomes smaller (but not larger) (Artelt & Hammer, 2022; Kenny & Keane, 2021), another modeling approach refers to semi-factuals as *minimum sufficient reasons (MSR)* (Barceló et al., 2020; Ignatiev & Marques-Silva, 2021), also called *Prime Implicant explanations*, aiming to identify the smallest subset of features that alone is sufficient for the observed outcome, i.e., ignoring the values the other features take, the outcome will always be unchanged:

$$\text{Does there exist an } \boldsymbol{x}_{\text{sf}} \text{ s.t. } h(\boldsymbol{x}_{\text{sf}}) = h(\boldsymbol{x}_{\text{orig}}) \text{ and } \|\boldsymbol{x}_{\text{sf}}\|_0 \leq k \text{ and } (\boldsymbol{x}_{\text{sf}})_i = (\boldsymbol{x}_{\text{orig}})_i \ \forall i : (\boldsymbol{x}_{\text{sf}})_i \neq \bot \tag{6}$$

where $\bot$ denotes the default (turned off) value of a feature. Like in the case of counterfactual explanations Eq. 5, Amir et al. (2024) extends Eq. 6 by an additional context indicator $\pi : \boldsymbol{x}_{\text{sf}} \mapsto \{0, 1\}$ for distinguishing between plausible ($\pi(\cdot) = 1$) and non-plausible ($\pi(\cdot) = 0$) semi-factual explanations:

$$\begin{aligned}
\text{Does there exist an } \boldsymbol{x}_{\text{sf}} \text{ s.t. } & h(\boldsymbol{x}_{\text{sf}}) = h(\boldsymbol{x}_{\text{orig}}) \\
& \pi(\boldsymbol{x}_{\text{sf}}) = 1 \\
& \|\boldsymbol{x}_{\text{sf}}\|_0 \leq k \\
& (\boldsymbol{x}_{\text{sf}})_i = (\boldsymbol{x}_{\text{orig}})_i \ \forall i : (\boldsymbol{x}_{\text{sf}})_i \neq \bot
\end{aligned} \tag{7}$$

Alternatively, Alfano et al. (2025) completely ignores the second objective and just maximizes the change such that the outcome remains unchanged:

$$\text{Does there exist an } \boldsymbol{x}_{\text{sf}} \text{ s.t. } h(\boldsymbol{x}_{\text{sf}}) = h(\boldsymbol{x}_{\text{orig}}) \text{ and } \theta(\boldsymbol{x}_{\text{orig}}, \boldsymbol{x}_{\text{sf}}) \geq k \tag{8}$$

where we abuse the previous notation and assume that $\theta(\cdot, \cdot)$ measures the distance/similarity of two given instances. Note that Eq. 8 assumes *binary/categorical* features in order to be meaningful (Alfano et al., 2025) and is also referred to as the *Maximum Change Allowed (MCA)*. Further note that only Eq. 6 and Eq. 8 have been considered when studying the computational complexity of semi-factual explanations.

## 2.2 Computational Complexity

In the following, we briefly introduce computational complexity and refer the interested reader to (Arora & Barak, 2009) for more details and formal definitions. There are different types of problems, and we will start by considering so-called *decision problems* where the objective is to compute a "yes" or "no" answer given some question on some input. As an example, Eq. 4 shows a formal way of stating the MCR problem as a decision problem in which the informal version of the related counterfactual question is as follows: Is it possible to change the class outputted by a classifier by applying a small change to the input?

Regarding the relation of two (decision) problems, we say that a problem A is *reducible* to a problem B if we can solve problem A using a subroutine for solving B without spending too much time translating A into B. In this case, we will write $A \leq B$ since B, in a sense, is at least as hard to compute as A.

Computational complexity classes state how difficult it is to solve a given (decision) problem. In this context, we say that an algorithm (for solving a given problem) is a polynomial time algorithm if the execution time is polynomial in the size of the input. Some of the fundamental complexity classes for decision problems, as relevant for this work, are the following:

- P (or PTIME): The class P contains all decision problems where there exist a polynomial time algorithm for computing the solution.

- NP, coNP and $D^p$: Informally speaking, a decision problem is in NP if there is an efficient procedure to prove that the answer is "yes" whenever this is the case. It is obvious that $P \subseteq NP$. It is an open problem whether $P \neq NP$, but the general belief is that this is true (Arora & Barak, 2009). A problem B is said to be NP-hard if $A \leq B$ holds for any problem A in NP. The relation $\leq$ is transitive, so we can show that a problem C is NP-hard by showing $B \leq C$ for a problem B known to be NP-hard. If we can show that a problem is NP-hard then it is unlikely that there is a polynomial time algorithm for solving it since this would imply $P = NP$. It is important to stress that a problem does not need to be a decision problem to be NP-hard. If a problem is NP-hard and a member of NP, we refer to the problem as being NP-complete. The coNP class contains the decision problems where "no" answers are efficiently verifiable. A decision problem is a member of the complexity class $D^p$ if the answer for some input is "yes" if and only if the answer is "yes" for some problem A and "yes" for some problem B on the same input where A and B are members of NP and coNP, respectively.

- $\Sigma_2^p$: Decision problems in $\Sigma_2^p$ also allow a formal procedure for proving any "yes" decision, but proofs may take superpolynomial time. Analogously to NP, a problem in $\Sigma_2^p$ can be $\Sigma_2^p$-complete, showing that the problem is at least as hard as any other problem in $\Sigma_2^p$. It is often conjectured that NP is a proper subset of $\Sigma_2^p$.

A second type of problems are *counting problems*, where the related question asks how many objects there are of a certain kind. As an example, let us revisit the MCR problem Eq. 4 for a classifier taking binary input. A counting version of MCR could be formulated as follows: How many inputs within Hamming distance 5 from the actual input can change the output of the classifier? Note that this problem is at least as hard as deciding if such an input exists. The class #P for counting problems is the analogous class to NP for decision problems, and it is unlikely that there is a polynomial time algorithm for solving a problem that is #P-complete since this would also imply $P = NP$. Occasionally, the problem of *enumerating* objects of a certain kind is considered – for example, explanations for ML models – instead of counting them.

The third and final type of problems that we consider in this work are *optimization problems*, where we focus on the subclass of *minimization problems*. Let us again consider the MCR problem Eq. 4, where a minimization version could be stated as follows: What is the closest input to the actual input for which the output of the classifier will change? Here we are trying to apply a minimum change to the input to change the output. In many cases, it is hard to find an exact solution to minimization problems, so we are looking for approximate solutions instead. An algorithm is a *c-approximation algorithm* for a minimization problem if the algorithm is able to compute a solution with a value of the objective that is not bigger than $c$ times the

optimal value. With respect to the MCR example, this means that the change the algorithm recommends is within a factor $c$ from the minimum change required. A common technique (also applied in this work) for showing that a polynomial-time algorithm for computing an approximate solution is unlikely is to show that the problem is NP-hard. As noted above, the existence of such an algorithm would then imply P = NP, which is believed to be false.

### 2.3 Specific models considered in literature

Besides "classic" machine learning models such as decision trees, tree ensembles, multi-layer perceptrons (MLPs) (e.g., ReLU networks, i.e., MLPs with ReLU activation functions), perceptrons, k-nearest neighbors (kNN), etc., the literature on computational complexity of counterfactuals also considers some "less popular" (but more general) models such as free binary decision diagrams and decision lists for Boolean functions. A free binary decision diagram (*FBDD*) is a rooted directed acyclic graph encoding a boolean decision function where the non-leaf nodes correspond to the binary variables and leaf nodes encode the decision output (Barceló et al., 2020). Note that decision trees (*DT*) are special instances of FBDDs where the graph is a tree. A Decision List (*DL*) constitutes an ordered list of IF-THEN rules whereby the condition statement is written as a logical conjunction over different features (Ignatiev & Marques-Silva, 2021).

## 3 Overview of Computational Complexity Results

Existing work on the computational complexity of counterfactual and semi-factual explanations has studied a variety of different classifiers with a special focus on perceptrons, ReLU networks, and FBDDs – see Figure 1 for an overview.

### 3.1 Counterfactual Explanations

Most research on the computational complexity of counterfactuals considers classic counterfactuals (Definition 1) without any additional constraints on plausibility, robustness, etc. Only very little work exists on robustness (w.r.t. model changes), plausibility, or global counterfactuals. We provide a summary of the complexity results, including two (straightforward) implications from the literature (Corollaries 1 and 2), in Table 1. Note that almost all existing work assumes discrete (i.e., binary) features, although some do not make any assumptions.

From Table 1 is becomes apparent that the computation of counterfactuals for ensembles of models is hard, and also for most single models, except monotonic classifiers and decision trees & diagrams. The effect of many additional properties, such as robustness to model changes and plausibility, on the computational complexity remains unknown. However, it seems to be the case that adding plausibility constraints also makes the computation hard. Observing those "negative" results raises the question of implications for the XAI community. In particular, how reasonable it is to aim for optimal explanations – we elaborate on this further in Section 5.

### 3.2 Semi-factual Explanations

Similar to the limited work on semi-factual explanations in general, existing work on the computational complexity of semi-factuals is also limited compared to counterfactuals (see Table 1). Alfano et al. (2025) explicitly studies the computational complexity of semi-factuals for a few classifiers (following the maximum change modeling approach Eq. 8). However, most of the existing work (Bassan et al., 2025; Marques-Silva et al., 2021; Ignatiev & Marques-Silva, 2021; Barceló et al., 2020) on the computational complexity of semi-factuals adopts the minimum sufficient reasons modeling approach Eq. 6, and states some complexity results, often as a byproduct of their study of counterfactual explanations. We provide a summary of the complexity results in Table 2. Note that in contrast to counterfactual explanations (see Table 1), the problem of

---

[1]The stated complexity classes in this table assume the worst-case of a ReLU network for implementing the context indicator $\pi(\cdot)$ – complexities for other functions can be found in Amir et al. (2024).

[2]For $k \geq 1$ and the $l_1$ norm – for the $l_2$-norm the complexity is PTIME (Barceló et al., 2025). For a discrete (binary) domain it is always NP-complete (Barceló et al., 2025)

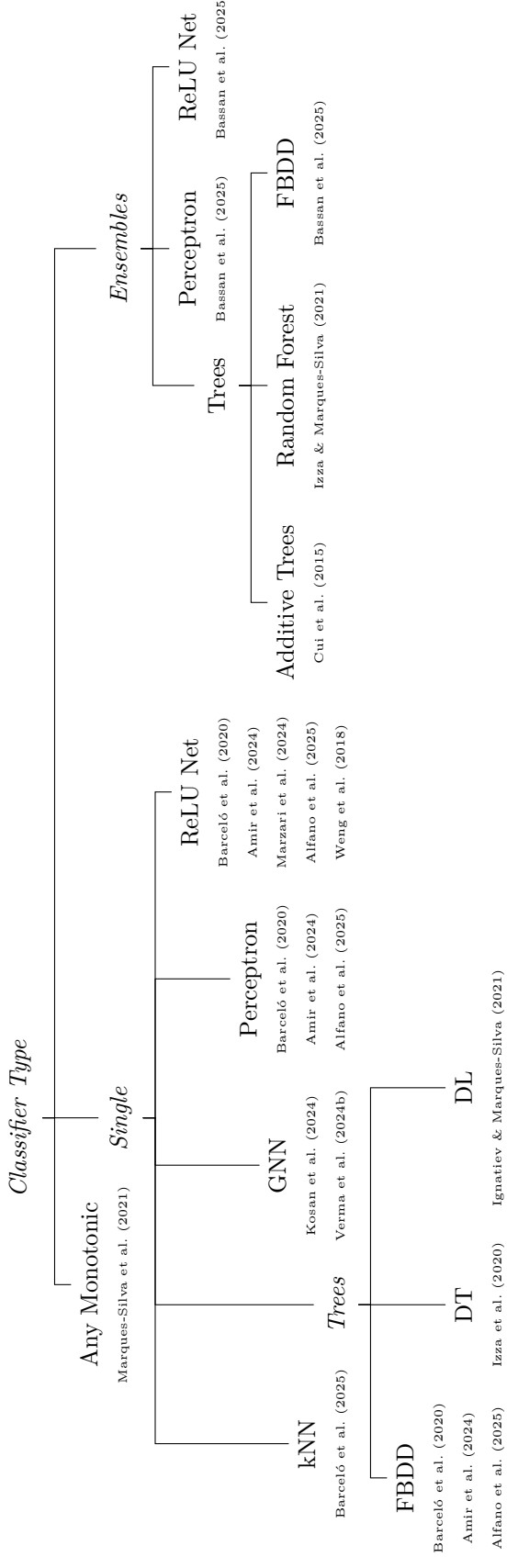

Figure 1: Overview of the classifier types for which the computational complexity of counterfactuals and/or semi-factuals has been studied in the literature.

| | Classifiers | Types of Counterfactuals | | | | |
|---|---|---|---|---|---|---|
| | | Classic (Single) | Classic (Enumerate) | Robust$_{Eq. 3}$ | Plausible$_{Eq. 5}$[1] | Global |
| Single | Any | - | NP-hard 
 Tsirtsis & Gomez Rodriguez (2020) | - | - | - |
| | Monotonic | PTIME 
 Marques-Silva et al. (2021) | NP-complete 
 Marques-Silva et al. (2021) | - | - | - |
| | kNN[2] | NP-complete 
 Barceló et al. (2025) | - | - | - | - |
| | GNN | NP-complete 
 Verma et al. (2024b) | - | - | - | NP-hard 
 Kosan et al. (2024) |
| | Perceptron | PTIME 
 Barceló et al. (2020) | - | - | NP-complete 
 Amir et al. (2024) | - |
| | ReLU Net | NP-complete 
 Barceló et al. (2020) | - | NP-hard 
 Marzari et al. (2024) | NP-complete 
 Amir et al. (2024) | - |
| Trees | DT | PTIME 
 Huang et al. (2021) | PTIME 
 Huang et al. (2021) | - | - | - |
| | FBDD | PTIME 
 Barceló et al. (2020) | - | - | NP-complete 
 Amir et al. (2024) | - |
| Ensembles | Perceptron | NP-complete 
 Bassan et al. (2025) | - | - | - | - |
| | ReLU Net | NP-complete 
 Bassan et al. (2025) | - | NP-hard 
 Corollary 1 | NP-complete 
 Corollary 2 | - |
| Trees | Additive Trees | NP-hard 
 Cui et al. (2015) | - | - | - | - |
| | FBDD | NP-complete 
 Bassan et al. (2025) | - | - | - | - |

Table 1: Computational complexity of counterfactual explanations.

| | | Modeling of Semi-factuals | | |
| | | Minimum Sufficient Reason (Eq. 6) | Plausible Eq. 7[3] | Maximum Change Allowed (Eq. 8) |
| | Classifiers | Classic | | Classic |
|---|---|---|---|---|
| Single | Monotonic | PTIME
Marques-Silva et al. (2021) | - | - |
| | kNN[4] | NP-hard
Barceló et al. (2025) | - | - |
| | Perceptron | PTIME
Barceló et al. (2020) | $\Sigma_2^p$-complete
Amir et al. (2024) | PTIME
Alfano et al. (2025) |
| | ReLU network | $\sum_2^p$-complete
Barceló et al. (2020) | $\Sigma_2^p$-complete
Amir et al. (2024) | NP-complete
Alfano et al. (2025) |
| | Extended Linear | PTIME
Marques-Silva et al. (2020) | - | - |
| Trees | DT | PTIME
Izza et al. (2020) | - | - |
| | DL | NP-hard
Ignatiev & Marques-Silva (2021) | - | - |
| | FBDD | NP-complete
Barceló et al. (2020) | $\Sigma_2^p$-complete
Amir et al. (2024) | PTIME
Alfano et al. (2025) |
| Ensembles | Perceptron | $\sum_2^p$-complete
Bassan et al. (2025) | - | - |
| | ReLU network | $\sum_2^p$-complete
Bassan et al. (2025) | - | - |
| | FBDD | $\sum_2^p$-complete
Bassan et al. (2025) | - | - |
| | Random Forest | $D^p$-complete
Izza & Marques-Silva (2021) | - | - |

Table 2: Computational complexity of semi-factual explanations.

enumerating all possible semi-factual explanations has not been studied in the literature. Only Marques-Silva et al. (2021) proves that enumerating all possible MSRs (Eq. 6) with a polynomial-time delay is NP-complete. Similar to the case of counterfactual explanations (see Section 3.1), Table 2 shows that the computation of semi-factuals for almost all models (except linear models and decision trees) is computationally hard.

## 4 New Inapproximability Results for Counterfactual Explanations

We identify a gap in the literature regarding the inapproximability of counterfactual and semi-factual explanations, as most of the literature considers only their exact computation. An exception is a result by Weng et al. (2018) showing that, unless NP = P, no polynomial time $(1 - o(1)) \ln n$-approximation algorithm exists for the MCR problem (Eq. 4) with $\theta(\cdot) = \ell_1(\cdot)$ for ReLU networks where $n$ is the number of neurons. This means that any polynomial time algorithm for computing the closest input that will change the output can face a scenario where the computed counterfactual is at a distance at least $\ln n$ times the optimal (roughly).

Focusing on counterfactual explanations, we extend this area by introducing new inapproximability results for ReLU neural networks, additive tree models and kNN models. Before detailing the inapproximability results, we revisit the counterfactual explanation problem as defined by Wachter et al. (2017). We refer to the optimization problem defined in Remark 1 as *WACHTER-CFE* and define its input and output as follows.

**Definition 2** (WACHTER-CFE problem)**.** *For a given regressor $h(\cdot)$, an instance $\boldsymbol{x}_{orig}$, and a requested prediction $y_{cf}$, the computation of a counterfactual explanation of the prediction $h(\boldsymbol{x}_{orig})$ is phrased as the following optimization problem (equivalent to Eq. 2):*

$$\arg\min_{\delta_{cf}} \ \ell(h(\boldsymbol{x}_{orig} \oplus \delta_{cf}), y_{cf}) + \lambda \cdot \theta(\delta_{cf}) \tag{9}$$

*where $\lambda > 0$ denotes a user-specified hyperparameter for balancing between obtaining the requested prediction $y_{cf}$ and minimizing the cost/complexity of the counterfactual $\delta_{cf}$.*

*The final counterfactual sample $\boldsymbol{x}_{cf}$ is obtained by applying the counterfactual $\delta_{cf}$ to the original instance $\boldsymbol{x}_{orig}$: $\boldsymbol{x}_{cf} = \boldsymbol{x}_{orig} \oplus \delta_{cf}$.*

Note that we extend the notion of $h(\cdot)$ as a classifier to the more general notion of a *regressor*. Since classification is a special case of regression, Definition 2 covers a larger spectrum of applications. Furthermore, Eq. 9 from Definition 2 can be reduced to the MCR formulation Eq. 4 by setting $\lambda$ to a sufficiently small positive number[5], recovering the constrained optimization problem where the cost (i.e., amount of change) has to be minimized subject to the correctness of the counterfactual $\delta_{cf}$. Therefore, our study of WACHTER-CFE (Definition 2) allows us to make more general statements, implicitly including the MCR formulation Eq. 4. The inputs $\boldsymbol{x}_{orig}$ are discrete/categorical inputs, for example, modeling yes or no features. While we could compute an exact solution to the WACHTER-CFE problem (Definition 2), we have already shown that this is known to be difficult in a variety of settings. Thus, an alternative would be to compute an approximate solution, i.e., building an efficient algorithm guaranteeing a solution with a value of the objective within some factor $K$ of the optimal value for some $K$ close to 1.

In the following, we investigate the computational complexity of such approximation algorithms for the WACHTER-CFE problem (Definition 2), and prove that such algorithms do not exist for classic regressors with discrete input, even if $K$ is *exponential* in the size of the regressors. For any polynomial $p(n)$, we show that there is no polynomial time algorithm for the WACHTER-CFE problem (Definition 2) with approximation factor $2^{p(n)}$ even for simple neural networks, additive tree models and kNN models with discrete input under the assumption P $\neq$ NP ($n$ refers to the size of the regressors, such as the number of neurons or number of nodes). More precisely, let $\delta_{cf}$ be the optimal value according to Definition 2. The presented theorems state that there does *not* exist any polynomial-time algorithm that is guaranteed to

---

[3]The stated complexity classes in this table assume the worst-case of a ReLU network for implementing the context indicator $\pi(\cdot)$; complexities for other functions can be found in Amir et al. (2024).

[4]For $k \geq 1$ and continuous or discrete (binary) domain – not matter which norm is used.

[5]The exact value depends on the specific problem and its parameters, such as the upper bound $k$ from Eq. 4.

compute a solution $\delta_{\text{cf}}'$ to Definition 2 such that:

$$\ell(h(\boldsymbol{x}_{\text{orig}} \oplus \delta_{\text{cf}}'), y_{\text{cf}}) + \lambda \cdot \theta(\delta_{\text{cf}}') \leq 2^{p(n)}(\ell(h(\boldsymbol{x}_{\text{orig}} \oplus \delta_{\text{cf}}), y_{\text{cf}}) + \lambda \cdot \theta(\delta_{\text{cf}})) \tag{10}$$

This means that computing approximately optimal counterfactuals is also computationally hard, and we can not hope to find an efficient (i.e., polynomial-time) approximation algorithm computing a $\delta_{\text{cf}}'$ satisfying Eq. 10.

The presented theorems hold for regressors $h(\cdot)$ taking binary input, i.e., $\mathcal{X} = \{0,1\}^d$ as it is usually assumed in the literature on computational complexity of counterfactual explanations (Amir et al., 2024). Note that practitioners often face discrete or binary/categorical features in applications, such as attrition and business analytics (Artelt & Gregoriades, 2023; 2024). Therefore, our presented theorems directly apply to important real-world scenarios and provide insights into how well optimal counterfactual explanations can be approximated in such scenarios. The loss $\ell(\cdot)$ is the squared error, and the cost $\theta(\cdot)$ of the counterfactual is a function satisfying $\max_z \theta(z) \leq q(n)$ for some polynomial $q(\cdot)$. As an example, the loss $\theta(\cdot)$ could be the 1-norm for the binary vector $\delta_{\text{cf}}$, and the operator $\oplus$ could be the xor operator, implying that we try to minimize the number of bits to flip in the input to get a good output from the regressor. The proofs of our results can be found in Appendix A.2, and are based on a 3-SAT reduction, which constitutes a common proof strategy for proving NP-completeness of a problem (Arora & Barak, 2009) and have also been used in some work on computational complexity of counterfactuals (Marzari et al., 2024). The formal statements of the theorems are as follows:

**Theorem 1.** *If P$\neq$NP, then the following holds for any polynomial $p(n)$: There is no polynomial time $2^{p(n)}$-approximation algorithm for the WACHTER-CFE problem (Definition 2) for neural networks (ReLU) with $n$ nodes and one hidden layer.*

For additive tree models, we will restrict our attention to forests consisting of binary decision trees where the aggregated response is the average of all the outputs of the trees.

**Theorem 2.** *If P$\neq$NP, then the following holds for any polynomial $p(n)$: There is no polynomial time $2^{p(n)}$-approximation algorithm for the WACHTER-CFE problem (Definition 2) for additive tree models with a total of $n$ nodes.*

Finally, we have a theorem for kNN models:

**Theorem 3.** *If P$\neq$NP, then the following holds for any polynomial $p(n)$: There is no polynomial time $2^{p(n)}$-approximation algorithm for the WACHTER-CFE problem (Definition 2) for kNN models of size $n$.*

## 5 Discussion of Open Questions & Implications for the XAI Community

Computational complexity analysis provides a critical view of the difficulty of computing various types of counterfactual and semi-factual explanations. This analysis can help users of XAI methods towards models that best suit the kinds of explanations they require. We now discuss the wider implications of computational complexity analysis, in the hopes of spurring more interest within the XAI community to analyze the complexity of existing and new methods.

We propose that computational complexity should be taken into consideration when proposing new explanation methods. While a new explanation type/formalization might have some appealing properties, it is also important to make sure that those explanations can be computed efficiently. Furthermore, it ought to be clearly understood which parameters (e.g., dimensionality, model-size, etc.) constitute the computational bottleneck.

As highlighted in this work, many counterfactual and semi-factual explanations are "hard" to compute. This might question their usefulness. That is, we must ask whether it makes sense to ask for such explanations if they are NP-hard/complete to compute, and also difficult to approximate. We argue that it might be reasonable to move away from the idea of computing the "best" (i.e., closest or optimal) counterfactual/semi-factual, and instead aim for some counterfactual/semi-factual with reasonable properties such that it still remains useful in practice. Indeed, existing work on computing counterfactual/semi-factual explanations

often utilizes heuristics that end up in local minima, yet those explanations often turn out to be useful (Smyth & Keane, 2022). In this context, we propose that future work should focus more on average-case performance analysis, as well as on less strict formulations that allow non-optimal solutions, such as Definition 2. This would better align with practice and bridge the gap between theoretical worst-case analyses and the scenarios in the real world.

Computational complexity analysis usually assumes categorical input features Amir et al. (2024). Although categorical features occur quite often in real-world scenarios, such as attrition and business analytics (Artelt & Gregoriades, 2023; 2024), there also exist scenarios with continuous features. In this context, the presented findings and complexity results do provide valuable insights for many real-world scenarios involving categorical features. In particular, they state that in those scenarios, even with popular state-of-the-art methods such as evolutionary methods (e.g, DiCE) (Mothilal et al., 2020; Dandl et al., 2020), we can not expect to get an optimal counterfactual/semi-factual explanation or even a "good" approximation. However, the presented findings do not provide statements for cases with continuous features, leaving them as a largely unexplored area from a computational complexity perspective. Existing work on continuous features often proves optimality by proving convexity of the optimization problem to be solved or using convex approximations to prove approximate optimality (Artelt & Hammer, 2020). However, formal statements for general non-convex scenarios remain an open problem, with the consequence that for many gradient-based methods, it remains unclear how optimal the generated explanations are.

A major question is whether it is acceptable to use classifiers/regressors for which the computation of (certain) explanations is "infeasible". We argue that this likely depends on the application domain and the importance of "optimal/best" counterfactual/semi-factual explanations. While in some applications a sufficiently good explanation will suffice (as discussed in the previous paragraph), other applications might require exact ones, with the consequence that not every classifier type can be used, given the computational hardness of computing such exact/optimal counterfactual/semi-factual explanations. In this context, our comprehensive overview of the computational complexity for different types of models and counterfactuals/semi-factuals can help decision makers in deciding whether it is appropriate to use counterfactual/semi-factual explanations in a particular application or not.

## 6   Conclusion

This work addressed the computational complexity of generating explanations for a variety of models and types of counter/semi-factual explanations. Our overview of the current state-of-the-art shows that there remain many open questions and thus opportunities for advancing current knowledge about the complexity of different explanations. We observed that in many settings, the computation of counterfactual and semi-factual explanations is computationally hard. We extended those findings further by showing that we cannot even efficiently compute counterfactuals with an exponential approximation factor.

Most existing work on explanations considers "plain" counterfactual and semi-factual explanations, which are known to miss/lack other important properties such as plausibility, robustness, etc. More research on the complexity of these extended definitions is needed, as outlined in the missing entries in Tables 1 and 2. In this context, it is worth noting that there exist numerous different formulations of robustness and plausibility, leaving a large field open for future research. Furthermore, the absence of any complexity results on causal counterfactual explanations is alarming given the fact that counterfactuals are often deployed in the real world, where actionability and causality are key requirements.

We acknowledge an important limitation of our work, namely that just because a particular type of explanation is shown to be computationally difficult, this does not mean that these types of explanations are useless or impractical in practice. Heuristic or approximation methods could be used in place of optimal approaches, but the compatibility of these methods with industrial requirements and regulatory schemes must be carefully considered. For future work, investigating the quality of heuristic and approximation schemes from a formal perspective could provide critical guidance for industry and governments for using and regulating XAI systems. In this context, there is also an urgent need for formal statements on the optimality of existing and commonly used heuristic and approximation methods for cases with continuous features, given that those cases have been mostly ignored in computational complexity analysis so far. It

would also be interesting to move away from worse-case analyses, towards average-case analyses, considering the WACHTER-CFE formulation instead of the strict MCR formulation for allowing non-optimal solutions, as is usually done in practice. Finally, the limited work on semi-factual explanations (compared to counterfactual explanations) is alarming and needs more attention from the community. Not only concerning their computational complexity, but also in general, such as evaluation methods, comparison of different formalization and modeling approaches, as well as algorithms for efficiently computing semi-factual explanations. In particular, there is an urgent need for (in-)approximability studies of semi-factuals, similar to what we did in Section 4. However, their difference from counterfactual explanations, in particular the absence of the contrasting property, hinders the transferability of existing results for counterfactuals to semi-factuals.

**Acknowledgments**

This research was supported by the Ministry of Culture and Science NRW (Germany) as part of the Lamarr Fellow Network. This publication reflects the views of the authors only.

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

## A  Appendix

### A.1  Counterfactual Explanations

#### A.1.1  Ensembles of ReLU Networks

**Corollary 1.** *The computation of robust counterfactual explanations (as defined in Marzari et al. (2024)) for an ensemble of ReLU networks is NP-hard.*

*Proof.* An ensemble of MLPs can be written as a single MLP – in particular for ReLU networks. Consequently, the statement follows from (Marzari et al., 2024). □

**Corollary 2.** *The computation of plausible counterfactual explanations (as defined in (Amir et al., 2024)) for an ensemble of ReLU networks is NP-complete.*

*Proof.* An ensemble of MLPs can be written as a single MLP – in particular for ReLU networks. Consequently, the statement follows from (Amir et al., 2024). □

### A.2  Proofs of Theorem 1, 2 and 3

For our three proofs, we will use a reduction from the famous 3-SAT problem that is NP-complete (Garey & Johnson, 1990). Note that reductions from 3-SAT constitute a popular proof strategy for proving NP-completeness of a problem (Arora & Barak, 2009). In the context, of the complexity analysis of counterfactuals and semi-factual explanations, reductions to existing problems constitute the most popular proof strategy. In particular, reductions from 3-SAT (Marzari et al., 2024), SAT (Ignatiev & Marques-Silva, 2021), vertex cover (Barceló et al., 2025), and Knapsack (Alfano et al., 2025). In our case, we aim for a reduction from 3-SAT because it aligns well with the contrasting property of counterfactuals and has been successfully used before as a proof strategy (Marzari et al., 2024). The input to the 3-SAT problem is a Boolean formula in conjunctive normal form (CNF) with clauses consisting of no more than three literals. The objective is

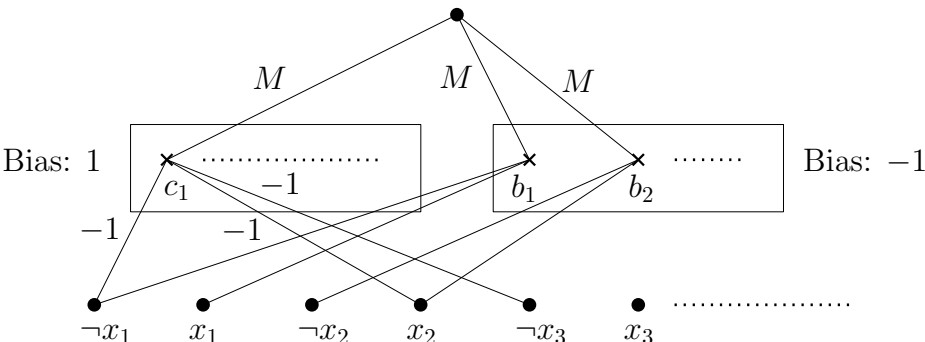

Figure 2: A neural network with one hidden layer used as the regressor $h(\cdot)$ in the reduction from the 3-SAT problem to the WACHTER-CFE problem (Definition 2). The clause $\neg x_1 \vee x_2 \vee \neg x_3$ is a clause in the CNF formula defining the 3-SAT instance. Connections with no weight shown in the figure have weight 1.

to decide whether the Boolean formula is satisfiable. Actually, we are using a restricted version of 3-SAT where each clause contains exactly three distinct literals. There is a straightforward, well-known reduction showing that the restricted version is NP-complete as well.

*Proof of Theorems 1:* As mentioned earlier, we will use reduction from the restricted 3-SAT problem. The intuition behind the proof is to turn an instance of 3-SAT into a WACHTER-CFE instance (Definition 2) with a regressor that outputs either 0 or a very big number $M$ where 0 corresponds to yes-instances of 3-SAT. If we efficiently can solve the WACHTER-CFE problem (Definition 2) approximately with target $y_{cf} = 0$, then we also can solve 3-SAT efficiently, which is a contradiction. Now, assume that there is a polynomial time $2^{p(n)}$-approximation algorithm $A(\cdot)$ for the WACHTER-CFE problem (Definition 2).

An instance of the 3-SAT problem is transformed into the WACHTER-CFE instance $(h, \boldsymbol{x}_{\text{orig}}, y_{cf}, \lambda)$ where $h(\cdot)$ is the neural network with one hidden layer shown in Fig. 2 with $M = \sqrt{\max_z \theta(z) \cdot 2^{p(n)} + 1}$ and $\boldsymbol{x}_{\text{orig}} = 0$, $y_{cf} = 0$, $\lambda = 1$. The regressor $h(\cdot)$ has an input neuron for each literal encoded according to the boolean value of the corresponding Boolean variable. For each clause, there is a neuron in the hidden layer that will have output 0 if the clause is satisfied and $M$ otherwise. There is also a neuron in the hidden layer for each Boolean variable that outputs $M$ if both the neurons for the literals for that variable in the input layer is 1 and 0 otherwise.

The number of nodes $n$ in the regressor $h(\cdot)$ is polynomial in the size of the 3-SAT instance, and $\log M$ is polynomial in $n$. This leads to the following observation:

**Observation 1.** *The WACHTER-CFE instance can be built in polynomial time.*

Another key observation is the following, which is not difficult to prove:

**Observation 2.** *The output of the regressor $h(\cdot)$ is either 0 or at least $M$. If $\boldsymbol{x}$ is a vector corresponding to a satisfying assignment of the Boolean variables for the 3-SAT instance, then $h(\boldsymbol{x}) = 0$. If $h(\boldsymbol{x}) = 0$, then we can construct a satisfying assignment of the Boolean variables for the 3-SAT instance from $\boldsymbol{x}$.*

Let $\boldsymbol{x}_{cf} = A(h, \boldsymbol{x}_{\text{orig}}, y_{cf}, \lambda)$ be the output of the approximation algorithm $A(\cdot)$ for computing a counterfactual explanation. Now, assume that the 3-SAT instance is a yes-instance. Let $\boldsymbol{x}_{3SAT}$ be the vector for an assignment of the Boolean variables satisfying the 3-SAT instance. Based on Observation 2, we now have $h(\boldsymbol{x}_{3SAT}) = 0$ implying the following inequality where $f(\cdot)$ denotes the WACHTER-CFE objective and $OPT_{CFE}$ denotes an optimal value for $f(\cdot)$:

$$OPT_{CFE} \leq f(\boldsymbol{x}_{3SAT}) \leq \max_z \theta(z) \tag{11}$$

The algorithm $A(\cdot)$ is a $2^{p(n)}$-approximation algorithm:

$$f(\boldsymbol{x}_{cf}) \leq 2^{p(n)} \cdot OPT_{CFE} \tag{12}$$

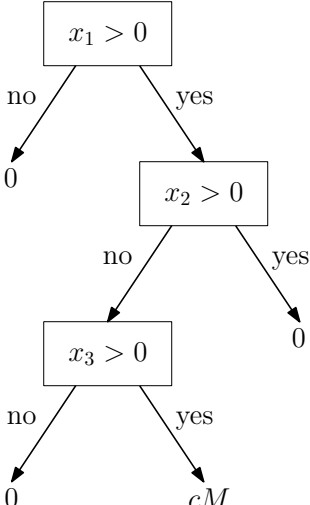

Figure 3: For each clause in the 3-SAT instance, we have a decision tree in the additive tree model producing a high output if the clause is not satisfied. The figure shows a decision tree for the example clause $\neg x_1 \vee x_2 \vee \neg x_3$.

$$\left[0, 1, 0, \tfrac{1}{2}, \tfrac{1}{2}, \tfrac{1}{2}, \tfrac{1}{2}, \ldots\right] : 0$$

$$\left[1, 1, 0, \tfrac{1}{2}, \tfrac{1}{2}, \tfrac{1}{2}, \tfrac{1}{2}, \ldots\right] : 0 \qquad \left[0, 0, 0, \tfrac{1}{2}, \tfrac{1}{2}, \tfrac{1}{2}, \tfrac{1}{2}, \ldots\right] : 0 \qquad \left[0, 1, 1, \tfrac{1}{2}, \tfrac{1}{2}, \tfrac{1}{2}, \tfrac{1}{2}, \ldots\right] : 0$$

$$\left[0, 0, 1, \tfrac{1}{2}, \tfrac{1}{2}, \tfrac{1}{2}, \tfrac{1}{2}, \ldots\right] : 0 \qquad \left[1, 1, 1, \tfrac{1}{2}, \tfrac{1}{2}, \tfrac{1}{2}, \tfrac{1}{2}, \ldots\right] : 0 \qquad \left[1, 0, 0, \tfrac{1}{2}, \tfrac{1}{2}, \tfrac{1}{2}, \tfrac{1}{2}, \ldots\right] : 0$$

$$\left[1, 0, 1, \tfrac{1}{2}, \tfrac{1}{2}, \tfrac{1}{2}, \tfrac{1}{2}, \ldots\right] : cM$$

$$x' = [1, 0, 1, 1, 1, 0, 1, \ldots]$$

Figure 4: The figure shows the 8 vectors in the kNN-regressor for the clause $\neg x_1 \vee x_2 \vee \neg x_3$ with labels after the colon symbol. The input vector $x'$ does not satisfy the clause, so it is closest to the vector with label $cM$.

We now get

$$f(\boldsymbol{x}_{\mathrm{cf}}) \leq 2^{p(n)} \cdot \max_z \theta(z) = (M-1)^2 \tag{13}$$

From Observation 2, we see that this inequality can only hold if $h(\boldsymbol{x}_{\mathrm{cf}}) = 0$ and that we can build a satisfying assignment for the 3-SAT instance from $\boldsymbol{x}_{\mathrm{cf}}$. In other words, the algorithm $A(\cdot)$ can be used to compute an assignment of the boolean variables satisfying the 3-SAT instance if such an assignment exists contradicting NP$\neq$P (here we use Observation 1). $\qquad\square$

*Proof of Theorem 2:* We use the same strategy as in the proof of Theorem 1. We construct a simple additive tree model with a similar functionality as the neural network.

We build a binary decision tree for each clause that will have output 0 if the clause is satisfied and output $cM$ otherwise where $c$ is the number of clauses. An example of such a tree is shown in Fig 3. Observation 1 and Observation 2 also hold for the additive tree model reduction, which concludes the proof. $\qquad\square$

*Proof of Theorem 3:* For a 3-SAT instance, we construct a simple kNN model with a similar functionality as the regressors in the previous two reductions. To be more specific, we build a kNN-regressor consisting of $8c$ vectors ($c$ is the number of clauses). The output of the regressor is the average of the labels of the $c$ vectors that are closest to the input with respect to the Euclidean distance.

For each clause, we construct 8 vectors corresponding to all the 8 possible assignments for the 3 boolean variables appearing in the clause. One of the vectors corresponds to the case where the clause is not satisfied, and this vector has the label $cM$, while the other 7 vectors have the label 0. The 8 vectors for an example clause are shown in Fig. 4.

The input vector $x'$ is not satisfying the clause *if and only if* the vector with label $cM$ is closest to $x'$, implying that Observation 2 holds. We also notice that Observation 1 holds, which concludes the proof. $\square$

