# OpenReview forum: "On the Hardness of Computing Counterfactual and Semi-factual Explanations in XAI"
_TMLR — Accepted by TMLR_

### Review · Reviewer_VsYR · 2025-06-11

**Summary Of Contributions:**

In this paper, the authors try to:
* provide an overview of the literature analyzing the computational complexity of counterfactual and semi-factual explanations.
* investigate the computational complexity of approximate solutions for counterfactual explanations. The authors conclude that explanations are often hard to generate and, under certain assumptions, they are also hard to approximate.

**Audience:**

Yes

**Broader Impact Concerns:**

No concerns regarding ethical implications.

**Claims And Evidence:**

Yes

**Requested Changes:**

* The authors state that _"This work focuses solely on this non-causal approach because all of the literature on the computational complexity of counterfactual explanations currently only considers the non-causal case."_ However, could the authors provide any insights and/or enrich the discussion regarding the computational complexity of causal counterfactuals?

* There are some sentences that should be rephrased and/or explained in a clearer way. For instance, in page 8 the authors state that _"no polynomial time (1 − o(1)) ln n-approximation algorithm exists for the MCR problem Eq. 3 for ReLU networks where n is the number of neurons unless NP = P."_

* The paper mentions robust and plausible counterfactual explanations several times, including in the appendices, but I think it never defines what is exactly meant by robustness or by plausibility. I know that the authors directly refer to the definitions by Marzari et al. (2024) and Amir et al. (2024) but, in order to make the paper a bit more self-contained in this aspect, I would try to present the concepts in the paper.

* Typos:
   * "also known as as “even if” explanations" $\rightarrow$ "also known as “even if” explanations"
   * the MSR acronym is defined twice: in page 3 and in page 7; the PI acronym is defined in page 3 but never used; the MCA acronym is defined in page 3 but never used; the acronyms FBDD and DL are defined twice in page 5; acronym kNN is presented in page 9 ("k-nearest neighbors (kNN)") when it has already been used 4 times. I guess CFE stands for "CounterFactual Explanation", but this should be clarified in the paper as well.
   * footnote 1 in page 7, and footnote 3 in page 8 are (almost) the same... Why? Furthermore, both footnotes include the acronym "FBDNN" that has not been defined before.
   * in page 9, if I am not mistaken, the hyperlink to Eq.2 is broken in "Objective: minimize objective (2)"

**Strengths And Weaknesses:**

Strengths:
* I believe that the study of the computational complexity of explainability methods in AI (and xAI in general) is a highly relevant topic with current impact.
* The fact that this paper addresses not only counterfactual explanations but also semi-factual ones is, in my view, very positive (since semi-factuals represent a less explored area).

Weaknesses:
* In my humble opinion, the paper lacks a bit of depth and original contribution. It starts by discussing explainable AI, focusing on counterfactuals and semi-factuals; then it shifts, in a generic, introductory-tutorial style, to computational complexity classes; afterwards, it gives an overview of the literature findings on the computational complexity of counterfactual and semi-factual explanations. And up to this point, we have covered eight pages (out of the ten pages of the paper). In other words, up to this point we could say that no new knowledge has been generated at all.
* Related with the previous point, the explanation of computational complexity classes (Section 2.2) seems a bit generic and unnecessary. For example, it discusses, among others, the coNP class, which is not mentioned or addressed again throughout the manuscript.
* The conclusions of the paper are somewhat disappointing. For instance, one of the main points is: _"We argue that it might be reasonable to move away from the idea to compute the 'best' (i.e., closest or optimal) counterfactual/semifactual, and instead aim for some counterfactual/semi-factual with reasonable properties such that it still remains useful in practice."_ This conclusion does not seem particularly novel or original. And I would argue that, in practical terms, this is already how counterfactuals are handled: you do not seek the optimal counterfactual, but one that is feasible to compute and useful for the specific practical case at hand.

---

> ### Author Response · Authors · 2025-11-03
> **Response to review**
>
> We thank the reviewer for their time and the detailed review. We appreciate that the reviewer considers the "study of the computational complexity of explainability methods in AI (and xAI in general) as a highly relevant topic with current impact.".
>
> We revised the paper to address all concerns and comments -- to facilitate the rebuttal process, we highlight all major changes (except typos) in blue.
> Below, we provide a detailed response to the reviewer's concerns and comments.
>
> >In my humble opinion, the paper lacks a bit of depth and original contribution. It starts by discussing explainable AI, focusing on counterfactuals and semi-factuals; then it shifts, in a generic, introductory-tutorial style, to computational complexity classes; afterwards, it gives an overview of the literature findings on the computational complexity of counterfactual and semi-factual explanations. And up to this point, we have covered eight pages (out of the ten pages of the paper). In other words, up to this point we could say that no new knowledge has been generated at all.
>
> As it is the nature of a survey/overview paper, we indeed did not develop any new algorithms. However, we argue that by providing a categorization of existing work on a certain topic and presenting it in a structured overview to highlight what already exists (or is known) and analyze what is missing can constitute a valuable contribution as well. Based on this, other researchers can easily identify research gaps and get a better understanding of the status of a field.
> Furthermore, we want to highlight that in addition to our categorization and structured overview, we do provide new algorithmic knowledge by stating several novel inapproximability results, which, however, are not the sole focus of the paper.
>
> >Related with the previous point, the explanation of computational complexity classes (Section 2.2) seems a bit generic and unnecessary. For example, it discusses, among others, the coNP class, which is not mentioned or addressed again throughout the manuscript.
>
> We included a brief overview of complexity classes to make the paper more accessible to readers who are less familiar with computational complexity theory.
> While coNP itself is not needed for any of the stated complexity results, the coNP class is needed to understand the definition of $D^p$ -- as stated in the paper, the class $D^p$ contains problems that are in the intersection of NP and coNP.
> We are open to shortening this section of the paper.
>
> >The conclusions of the paper are somewhat disappointing. For instance, one of the main points is: "We argue that it might be reasonable to move away from the idea to compute the 'best' (i.e., closest or optimal) counterfactual/semifactual, and instead aim for some counterfactual/semi-factual with reasonable properties such that it still remains useful in practice." This conclusion does not seem particularly novel or original. And I would argue that, in practical terms, this is already how counterfactuals are handled: you do not seek the optimal counterfactual, but one that is feasible to compute and useful for the specific practical case at hand.
>
> We agree with the reviewer that many toolboxes indeed just compute some feasible and practical counterfactual without worrying too much about optimality.
> However, we think it is still useful from a scientific point of view to better understand the complexity of the underlying problem that is being solved when computing counterfactual/semi-factual explanations. Furthermore, it sets boundaries for how "optimal" we can expect counterfactuals/semi-factuals to be, and it clarifies in which cases we can indeed guarantee optimality. This can be of particular importance in critical applications where optimality might be required. Having this knowledge at hand, stakeholders can then make an informed decision whether it is appropriate to use counterfactual/semi-factual explanations in a particular application or not.
>
> In the revised paper, we revised the last paragraph in section 5 to better highlight these practical benefits of having a comprehensive overview of the complexity of different models and types of explanations.

---

> > ### Author Response · Authors · 2025-11-03
> > **Response to review -- part 2**
> >
> > >The authors state that "This work focuses solely on this non-causal approach because all of the literature on the computational complexity of counterfactual explanations currently only considers the non-causal case." However, could the authors provide any insights and/or enrich the discussion regarding the computational complexity of causal counterfactuals?
> >
> > As stated in our paper, there does not exist any work studying the computational complexity of causal counterfactual explanations. It is therefore difficult to make general statements. In particular, there does not exist a globally agreed-upon formalization of causal counterfactuals. Usually, causal counterfactuals are informally related to counterfactuals where the set of possible actions is limited to a feasible set of actions derived from domain knowledge, or where the actions must adhere to some given (structural) causal model. Consequently, there exists a wide variety of different methods and formalizations that aim for causal counterfactuals.
> > However, for some special formalizations of causal counterfactuals, it is not too difficult to analyze their computational complexity.
> > For instance, the method called *actionable recourse* proposed by Ustun et al. (2019) focuses on linear classifiers and formalizes a causal counterfactual explanation as an integer optimization problem. Because integer optimization problems are NP-hard, so is their causal counterfactual method.
> > Similarly, the popular MACE (Model-Agnostic Counterfactual Explanations for Consequential Decisions) method for causal counterfactual explanations proposed by Amir et al. (2020), can also be analyzed easily. Here, a causal counterfactual explanation is computed by repeatedly querying an SMT oracle (i.e., solving a generalized SAT problem). Because SAT problems are already NP-complete, SMT problems are usually NP-hard or can even be undecidable. Consequently, the computation of such causal counterfactuals can also be flagged as NP-hard.
> > Note that although being NP-hard, those problems might be solvable in low dimensions or can be approximated "sufficiently" well at the cost of losing optimality. Furthermore, both approaches constitute special methods, making assumptions about the type of model and the underlying causal model. The computational complexity of general causal counterfactuals remains an open research question on its own.
> >
> > In the revised paper (in the conclusion), we highlight the need and importance of complexity results on causal counterfactuals. We are open to including a more detailed discussion of those cases in the paper.
> >
> > >There are some sentences that should be rephrased and/or explained in a clearer way. For instance, in page 8 the authors state that "no polynomial time (1 − o(1)) ln n-approximation algorithm exists for the MCR problem Eq. 3 for ReLU networks where n is the number of neurons unless NP = P."
> >
> > Thanks for pointing this out. In the revised paper, we added an explanation to enhance the clarity. In this context, we also added some additional clarifications to our own theorems, easing the reader's understanding.
> >
> > >The paper mentions robust and plausible counterfactual explanations several times, including in the appendices, but I think it never defines what is exactly meant by robustness or by plausibility. I know that the authors directly refer to the definitions by Marzari et al. (2024) and Amir et al. (2024) but, in order to make the paper a bit more self-contained in this aspect, I would try to present the concepts in the paper.
> >
> > Thanks for pointing this out.
> > In the revised paper, we added a paragraph about the plausibility modeling by Amir et al. (2024) for counterfactual and semi-factual explanations -- see Eq. 5 and 7. Basically, Amir et al. (2024) propose to use a context indicator (i.e., an additional function) for distinguishing between plausible and non-pausible counterfactuals/semifactuals. This context indicator function is then added to the optimization problem to enforce the plausibility of the final counterfactual/semifactual. Depending on the implementation of this context indicator function, the computational complexity of computing counterfactuals/semifactuals can change.
> >
> > Similarly, we also added a paragraph describing the robustness definition by Marzari et al. (2024) -- see Eq. 3. Note that Marzari et al. (2024) only focus on robustness with respect to model changes.

---

> > > ### Author Response · Authors · 2025-11-03
> > > **Response to review -- part 3**
> > >
> > > >Typos:
> > > >"also known as as “even if” explanations" "also known as “even if” explanations"
> > > >the MSR acronym is defined twice: in page 3 and in page 7; the PI acronym is defined in page 3 >but never used; the MCA acronym is defined in page 3 but never used; the acronyms FBDD and DL >are defined twice in page 5; acronym kNN is presented in page 9 ("k-nearest neighbors (kNN)") > when it has already been used 4 times. I guess CFE stands for "CounterFactual Explanation", but >this should be clarified in the paper as well.
> > > >in page 9, if I am not mistaken, the hyperlink to Eq.2 is broken in "Objective: minimize objective (2)"
> > >
> > > Thanks for pointing out those typos. We have fixed them all in the revised paper.
> > >
> > > >footnote 1 in page 7, and footnote 3 in page 8 are (almost) the same... Why? Furthermore, both footnotes include the acronym "FBDNN" that has not been defined before.
> > >
> > > Thanks for bringing this up. "FBDNN" is indeed another typo; it should be "FBDD" instead. Thanks for pointing this out.
> > > The two table footnotes are very similar because they both refer to the same concept of a *context indicator* [Amir et al. 2024] for ensuring the plausibility of counterfactuals and semi-factuals. In the revised paper, we revised those footnotes in the light of the added paragraphs about the context indicator function.

---

> > > > ### Comment · Reviewer_VsYR · 2025-11-03
> > > >
> > > > Dear all,
> > > >
> > > > I have reviewed the other reviewers' comments and the authors' responses (both to my comments and to those of the other reviewers). Actually, I maintain my view that "the paper lacks a bit of depth and original contribution [...and...] The conclusions of the paper are somewhat disappointing". But, if the other reviewers are inclined to accept the paper, I will not oppose it. Furthermore, TMLR emphasizes technical correctness over subjective significance (as stated in the website, "Papers should be accepted if they meet the criteria, even if the contribution or significance of the work is modest.").
> > > >
> > > > I find the authors' responses acceptable. And I believe the modifications included in the new version of the paper make it stronger.
> > > >
> > > > Best

---

> > > > > ### Comment · Action_Editor_nC6z · 2025-11-03
> > > > > **Let's engage in productive discussion please**
> > > > >
> > > > > Hi,
> > > > >
> > > > > Thank you for being on top of things and reviewing the author responses to your (and others') review! However, let's engage in discussion at this phase rather than jumping directly to a final conclusion of the work.
> > > > >
> > > > > Are there any aspects of the author response that you feel warrant further clarification? Are there parts of the work that you feel could be remediated to change your overall perspective?
> > > > >
> > > > > Let's give the authors the best opportunity to improve their work as far as is reasonable within the review timelines of TMLR. If not, let's give them specific feedback that would allow them to confidently revise their work for a potential resubmission.

---

> > > > > > ### Comment · Reviewer_VsYR · 2025-11-03
> > > > > >
> > > > > > Hi,
> > > > > >
> > > > > > Thank you for encouraging discussion and promoting the best possible feedback for the authors.
> > > > > >
> > > > > > In my case, I reviewed this paper within the stipulated time limit and provided my comments on June 11th (almost 5 months ago). I don't believe I have time right now to give more detailed feedback than I already have. I'm currently in one of the busiest periods of my academic year, and would need several days/weeks to find the suitable time to provide additional feedback or to debate in depth specific details of the paper.
> > > > > >
> > > > > > Best

---

> > > > > ### Author Response · Authors · 2025-11-04
> > > > >
> > > > > Many thanks for your quick response.
> > > > > Let us know if there is anything else we can clarify.
> > > > >
> > > > > Best wishes,
> > > > > The authors

---

### Review · Reviewer_gJzN · 2025-06-25

**Summary Of Contributions:**

This paper examines the computational complexity of generating counterfactual and semi-factual explanations. It begins with a concise introduction to both types of explanations and fundamental notions in computational complexity. It then surveys known results on the computational hardness of counterfactual generation and semi-factual generations across various model classes. From this setup, the authors derive two straightforward corollaries.

While prior work primarily focuses on exact solutions, this paper extends the discussion to approximation complexity: the difficulty of approximating a counterfactual explanation, rather than computing it exactly. To this end, the authors construct hardness proofs by reducing the counterfactual approximation problem to 3-SAT for three model classes (MLPs, regression trees, and kNNs). Each construction defines models that output 0 if the 3-SAT instance is satisfiable, and a large value otherwise. The approximation threshold is engineered so that any solution outside a narrow band would be too far from the optimal value, thus would not lead to a solution to the 3-SAT problem.

**Audience:**

Yes

**Claims And Evidence:**

Yes

**Requested Changes:**

- Clarify the scope and assumptions of the inapproximability theorems—particularly discrete inputs—and consider briefly how these assumptions compare to real-world counterfactual setups.
- Expand Section 5 to include a discussion of commonly used heuristic methods and how they fare in light of the intractability results.
- Discuss the originality of the 3-SAT reduction strategy by comparing it to similar constructions in existing literature. For instance, briefly introducing similar constructions in other papers in the literature review would help understand the novelty of the paper.

**Strengths And Weaknesses:**

# Strengths
- The paper is well written and accessible, especially in its introduction and background. It provides adequate context and formalisation. Despite my limited background in complexity theory, I was able to follow most of the arguments.
- The survey of existing work is extensive and well organised in tables.
- To the best of my knowledge, the approximation hardness results are novel and well constructed, demonstrating that approximate counterfactual generation is also computationally intractable.


# Weaknesses :
- The readability of the inapproximability proofs could be improved. Furthermore, in Definition 2 (WACHTER-CFE), it would help to include the full objective function and define the function $f$ explicitly. Including more of the intuition for the construction in the main text, not just in the appendix, would make the results more accessible.
- The 3-SAT reductions are interesting, but their novelty is hard to assess without a comparison to similar constructions in the literature. A brief discussion of related reduction techniques—especially in the context of approximation—would clarify the paper’s contribution.
- Since optimal solutions are shown to be hard or inapproximable, a survey or commentary on heuristic or approximate methods in the literature would be a helpful complement to the hardness results.
- The results are limited to discrete input domains.  It narrows the generality of the findings and the discussion in section 5. The implications for continuous domains (which are typical in many practical counterfactual generation setups) should be discussed further, especially in Section 5.
- The authors raise the question of whether it is acceptable to use models for which computing explanations is infeasible. However, many counterfactual methods already rely on heuristics (e.g., gradient descent, adversarial losses) that produce approximate explanations. It would be helpful to relate the theoretical results to this practical reality.
- A significant portion of the paper discusses semi-factual explanations, yet the main technical results pertain only to counterfactuals. Given the close relationship between the two (as noted in Alfano et al., 2025), the authors could do more to explore whether their techniques extend to semi-factuals.

---

> ### Author Response · Authors · 2025-11-03
> **Response to review**
>
> We thank the reviewer for their time and the detailed review.
>
> We revised the paper to address all concerns and comments -- to facilitate the rebuttal process, we highlight all major changes (except typos) in blue.
> Below, we provide a detailed response to the reviewer's concerns and comments.
>
> >The readability of the inapproximability proofs could be improved. Furthermore, in Definition 2 (WACHTER-CFE), it would help to include the full objective function and define the function
>  explicitly. Including more of the intuition for the construction in the main text, not just in the appendix, would make the results more accessible.
>
> Thanks for bringing this up. We completely revised Definition 2 to make it more accessible and improve overall clarity.
>
> >The results are limited to discrete input domains. It narrows the generality of the findings and the discussion in section 5. The implications for continuous domains (which are typical in many practical counterfactual generation setups) should be discussed further, especially in Section 5.
>
> Indeed, the results are limited to discrete (i.e., binary) input domains, as computational complexity analysis usually studies discrete inputs only. However, practitioners often face discrete (categorical) input spaces in real-world scenarios such as attrition or business analytics (e.g., Artelt & Gregoriades 2023,2024). For those cases, the presented complexity results provide valuable insights into the complexity of providing explanations in those scenarios.
> However, the presented complexity findings do not provide statements for cases with continuous features,
> leaving them as a largely unexplored area from a computational complexity perspective. Existing work on continuous features often proves optimality by proving convexity of the optimization problem to be solved or using convex approximations to prove approximate optimality (e.g, Artelt & Hammer, 2020). However, formal statements for general non-convex scenarios remain an open problem.
>
> We added a detailed discussion of those aspects to Section 5.
>
> >The authors raise the question of whether it is acceptable to use models for which computing explanations is infeasible. However, many counterfactual methods already rely on heuristics (e.g., gradient descent, adversarial losses) that produce approximate explanations. It would be helpful to relate the theoretical results to this practical reality.
>
> Indeed, most existing methods for computing counterfactual explanations constitute heuristics as they can not guarantee optimality, sometimes not even feasibility. The presented complexity results state that in those scenarios, even with popular state-of-the-art methods such as evolutionary methods (e.g, DiCE) Mothilal et al. (2020); Dandl et al. (2020), we can not expect to get an optimal counterfactual/semi-factual explanation or even a “good” approximation.
>
> We added a brief discussion of existing heuristic methods for computing counterfactual explanations to Section 2.1, and elaborate on the consequences of the presented complexity results for existing heuristics in Section 5. In this context, we also discuss the implications for real-world applications and the generality of the findings concerning different feature type domains.

---

> > ### Author Response · Authors · 2025-11-03
> > **Response to review -- part 2**
> >
> > >A significant portion of the paper discusses semi-factual explanations, yet the main technical results pertain only to counterfactuals. Given the close relationship between the two (as noted in Alfano et al., 2025), the authors could do more to explore whether their techniques extend to semi-factuals.
> >
> > Indeed, counterfactual and semi-factual explanations are related to each other, but also different -- in particular, the contrasting property (i.e., the output of the model must change) of counterfactual explanations is missing in semi-factual explanations. Furthermore (and in contrast to counterfactuals), for semi-factuals there does not exist a single, globally agreed on, formalization or modeling. As stated in our survey, the minimum sufficient reason and maximum change allowed constitute the most popular modeling approach in literature. However, from a computational and algorithmic perspective, they are very different.
> >
> > In the context of our novel inapproximability results for counterfactual explanations, it is worth noting that those do not easily transfer to semi-factual explanations because they rely on the contrasting property (i.e., the output of the model must change), which is missing in semi-factual explanations.
> > We agree that there is a need for more research on semi-factual explanations, not only concerning their computational complexity but also in general, such as evaluation methods, comparison of different formalization and modeling approaches, as well as algorithms for efficiently computing semi-factual explanations.
> > In the revised paper, we added a discussion of those aspects and the necessity of future work on semi-factual explanations to the conclusion in Section 6.
> >
> > >Clarify the scope and assumptions of the inapproximability theorems—particularly discrete inputs—and consider briefly how these assumptions compare to real-world counterfactual setups.
> >
> > Thanks for bringing this up. We added a paragraph clearly stating the assumptions and their implications and applicability in practice.
> >
> > >Expand Section 5 to include a discussion of commonly used heuristic methods and how they fare in light of the intractability results.
> >
> > In the revised paper, we added a discussion on the consequences of the presented complexity results for such heuristics in Section 5.
> >
> > >Discuss the originality of the 3-SAT reduction strategy by comparing it to similar constructions in existing literature. For instance, briefly introducing similar constructions in other papers in the literature review would help understand the novelty of the paper.
> >
> > Reductions from 3-SAT constitute a popular proof strategy for proving NP-
> > completeness of a problem (e.g., Arora & Barak, 2009). In the case of our novel inapproximability statements, the novelty lies in the statement but not in the utilized proof strategy.
> >
> > In the context of complexity analyses of counterfactuals
> > and semi-factual explanations, reductions from existing problems constitute the most popular proof strategy. In particular, reductions from 3-SAT (e.g., Marzari et al., 2024), SAT (e.g, Ignatiev & Marques-Silva, 2021), vertex cover (e.g., Barceló et al., 2025), and Knapsack (e.g., Alfano et al., 2025). In our case, we chose a reduction from 3-SAT because it aligns well with the contrasting property of counterfactuals and has been successfully used before as a proof strategy (e.g, Marzari et al., 2024).
> >
> > In the revised paper, we added a brief discussion of this directly to Section 4 when introducing our Theorems. Furthermore, at the beginning of the proofs in Appendix A.2, we added a more in-depth discussion and comparison to other proof strategies, as well as a justification for why we chose 3-SAT.

---

> ### Comment · Action_Editor_nC6z · 2025-11-15
> **Please engage with the author response**
>
> Hi Reviewer gJzN,
>
> The authors have carefully responded to your review and have also provided a revision of their paper. I cannot accept any recommendation you make on the paper without seeing your good faith effort to engage with them in discussion.
>
> Best,
> Taylor

---

> > ### Comment · Reviewer_gJzN · 2025-11-20
> >
> > I thank the authors for their detailed and constructive rebuttal and for the revised manuscript. After reading the updated version, I am satisfied that the main concerns raised in my review have been mostly addressed.
> >
> > My main issue with the original submission was similar to what the other reviewers noted: the contribution was not clearly delineated, as the paper sat somewhat ambiguously between being a survey and a technical contribution (see reviewers RLGA and VsYR’s first weaknesses). The revised version resolves this by clarifying the context, structuring the contribution more explicitly, and improving readability in key sections.
> >
> > In particular:
> > - Definition 2 (WACHTER-CFE) has been rewritten to include the full formal objective and improve readability.
> > - Section 5 now contains a clearer explanation of why the results assume discrete inputs, how this relates to real-world settings with categorical data, and what remains open for continuous feature spaces.
> > - The revised paper clarifies why the current techniques do not directly extend to semi-factuals and adds commentary on the need for further work on that topic.
> > - Section 4 and the appendix now include a comparison with prior reduction styles in the literature and position the contribution relative to existing proof techniques.
> >
> >
> >
> > I still think the proof structure could be improved in the appendix, but I believe the rest of the paper now provides a useful and well-structured overview of complexity results in XAI, along with interesting inapproximability theorems. I recommend acceptance.

---

> > > ### Comment · Action_Editor_nC6z · 2025-11-21
> > > **Thank you!**
> > >
> > > Thanks for your response! Can you please submit a formal recommendation? You can find the button at the top of this page next to the "Official Comment" button.

---

### Review · Reviewer_RLgA · 2025-10-31

**Summary Of Contributions:**

This paper investigates the computational complexity of generating counterfactual and semi-factual explanations for machine learning models. The central finding is that for many common and complex models, finding these explanations is computationally hard (e.g., NP-hard), and often hard to even approximate. This has significant implications for the XAI community and the explainability of ML models in general.

The authors' contributions are twofold:

1. Survey of counterfactual and semi-factual explanations of existing methods: The paper first provides a unifying framework of known complexity results. It summarises that while finding explanations for simple models like Decision Trees is efficient (PTIME), the problem becomes computationally hard for more powerful models. This includes being NP-complete for counterfactuals on ReLU Networks and kNN, and even $\Sigma_{2}^{p}$-complete for plausible semi-factuals on these models.

2. Inapproximability Results: This is the novel aspect of the paper. The authors propose inapproximability proofs for counterfactual and semi-factual finding and show that not only is finding exact optimal counterfactuals hard, but even finding approximate solutions is computationally difficult. Specifically, they prove (Theorems 1, 2, and 3) that for ReLU networks, additive tree models, and kNN models, no polynomial-time approximation algorithm exists (under the assumption $P \ne NP$).

Typically I look into 4 factors for judging a paper: Soundness, clarity, impact and novelty. The paper is very good in terms of the first three, while being novel on the inapproximation proof. Leaning towards acceptance.

**Audience:**

Yes

**Broader Impact Concerns:**

No Broader Impact concerns.

**Claims And Evidence:**

Yes

**Requested Changes:**

Requested changes:

1. Explicitly Bridge the Classifier-Regressor Gap: The paper would be significantly strengthened if the authors, in Section 4, explicitly justify the shift to a regressor formulation for their new proofs. They should add a discussion on how the inapproximability of the WACHTER-CFE regressor problem translates to the "classic" CFE classifier problem (MCR, Eq. 3). For instance, can any MCR problem for a classifier be reduced to this regressor formulation, preserving the approximation hardness?

2. Clarify the Choice of WACHTER-CFE: A brief paragraph could be added to Section 4 explaining why the WACHTER-CFE formulation (Eq. 2) was chosen for the inapproximability proofs over the MCR formulation (Eq. 3). This would help integrate the new contribution with the surveyed literature more tightly.

3. Refine the Discussion on Practicality: The discussion in Section 5 could be expanded to be more actionable. Instead of just stating that heuristics are used, the authors could suggest that future work should move beyond worst-case complexity. They could propose analyzing the average-case complexity or parameterised complexity (e.g., complexity in terms of model depth rather than just total size) of these problems, which might better explain why heuristics are often "good enough" in practice.

4. Contextualise "Plausibility" and "Robustness": The paper rightly identifies that the complexity of "plausible" and "robust" explanations is a key open area. It would be beneficial to add a sentence acknowledging that the definitions used in the literature so far (e.g., "learning a separate function") are just a few of many possible formalisations. This would frame the "missing entries in Tables 1 and 2" as an even richer field for future work, covering other definitions like causal plausibility or manifold-based robustness.

**Strengths And Weaknesses:**

Strengths of the paper:

1. The problem of counterfactual and semi-factual estimation is of significant importance given the rise of large language models and data privacy concerning these models, and there exists a lack of interpretability surrounding these models. This paper makes an attempt to solve that, by grouping together all the existing models and stating their respective complexities (Tables 1 and 2). An impactful contribution.

2. The authors then introduce the problem on approximating these counterfactuals (semi-factuals) as opposed to finding the exact solutions, and demonstrate they can be hard to find. They prove this using a reduction from the 3-SAT problem. The authors construct regressors (e.g., a ReLU network, an additive tree model) that encode a 3-SAT instance, creating a large "cost gap" between satisfiable ("yes") and unsatisfiable ("no") instances. An efficient approximation algorithm could distinguish between these costs, thus solving 3-SAT, which is believed to be impossible. Proof is precise and accurate.

3. The paper is very well written and easy to follow, with the problem statement being very clear in terms of clarity and impact.

Weaknesses of the paper:

1. Disconnect Between Survey and Novel Results: The paper's primary survey and background (Sections 2 and 3) are focused on classifiers, where the goal is to change a class label (e.g., Eqns 1, 3). However, the paper's main novel contribution (Section 4) proves inapproximability for the WACHTER-CFE formulation (Eq. 2), which is explicitly defined for a regressor using a squared error loss function. The link between the hardness of approximating this regressor problem and the hardness of the original classifier problem (finding the minimum-cost change to flip a label) is not made explicit.

2. Ambiguity in Problem Formulations: The new proofs in Section 4 are for the single-objective WACHTER-CFE formulation (Eq. 2). Most of the surveyed literature, however, appears based on the constrained-optimization "MCR" formulation (Eq. 3). It is not immediately clear if the inapproximability results for the $\lambda$-balanced objective (Eq. 2) directly apply to the MCR decision problem (Eq. 3).

3. Not necessarily a weakness, but a consideration for future work in my opinion: The discussion (Section 5) rightly points out that NP-hard results mean optimal explanations are often infeasible, and practitioners must rely on heuristics. However, it doesn't deeply explore why these heuristics often find "useful" (if non-optimal) explanations. The paper focuses on worst-case complexity (proven via 3-SAT reductions) but doesn't discuss the gap between this and the average-case performance on real-world models. I believe that's one of the next steps of the paper.

---

> ### Author Response · Authors · 2025-11-03
> **Response to review**
>
> We thank the reviewer for their time and the detailed review.
>
> We revised the paper to address all concerns and comments -- to facilitate the rebuttal process, we highlight all major changes (except typos) in blue.
> Below, we provide a detailed response to the reviewer's concerns and comments.
>
> >Disconnect Between Survey and Novel Results: The paper's primary survey and background (Sections 2 and 3) are focused on classifiers, where the goal is to change a class label (e.g., Eqns 1, 3). However, the paper's main novel contribution (Section 4) proves inapproximability for the WACHTER-CFE formulation (Eq. 2), which is explicitly defined for a regressor using a squared error loss function. The link between the hardness of approximating this regressor problem and the hardness of the original classifier problem (finding the minimum-cost change to flip a label) is not made explicit.
>
> The WACHTER-CFE formulation for regression constitutes a more general problem than the MCR formulation for classifiers, since classification is a special case of regression. Most importantly, the WACHTER-CFE formulation can also be reduced to MCR formulation by setting $\lambda$ to a sufficiently small positive number, recovering the constrained optimization problem from the MCR formulation. Therefore, besides considering the approximability as a novel aspect, our analyses on the WACHTER-CFE formulation extend existing work that considers the MCR formulation for classification on two dimensions: Regression as a more general task than classification; relaxing the optimality constraint in the MCR formulation for allowing non-optimal explanation, as it is usually done in practice.
>
> In the revised manuscript, we added a clarification (in Section 4) on how the regression problem generalizes the classification problem (also see our response to your second comment), and how our inapproximability results relate to existing classifier problems.
>
> >Ambiguity in Problem Formulations: The new proofs in Section 4 are for the single-objective WACHTER-CFE formulation (Eq. 2). Most of the surveyed literature, however, appears based on the constrained-optimization "MCR" formulation (Eq. 3). It is not immediately clear if the inapproximability results for the
> -balanced objective (Eq. 2) directly apply to the MCR decision problem (Eq. 3).
>
> The WACHTER-CFE formulation leads to more general results than just considering the MCR formulation. Most importantly, the WACHTER-CFE formulation reduces to the MCR formulation if we set $\lambda$ to a sufficiently small positive number, recovering the constrained optimization problem from the MCR formulation. However, it is worth noting that most algorithms for computing counterfactual explanations in practice actually aim to solve the WACHTER-CFE formulation instead of the MCR problem. Therefore, by focusing on the WACHTER-CFE formulation, our complexity statements are more aligned with practice than statements focusing on the MCR problem only.
>
> In the revised manuscript (Section 4), we added a clarifying statement about this to the paragraph right after Definition 2.

---

> > ### Author Response · Authors · 2025-11-03
> > **Response to review -- part 2**
> >
> > >Not necessarily a weakness, but a consideration for future work in my opinion: The discussion (Section 5) rightly points out that NP-hard results mean optimal explanations are often infeasible, and practitioners must rely on heuristics. However, it doesn't deeply explore why these heuristics often find "useful" (if non-optimal) explanations. The paper focuses on worst-case complexity (proven via 3-SAT reductions) but doesn't discuss the gap between this and the average-case performance on real-world models. I believe that's one of the next steps of the paper.
> >
> > Thanks for the suggestion. This is a very good point and definitely a very interesting approach for future research. It also aligns well with our study of the WACHTER-CFE problem, where we included the regularization coefficient $\lambda$, for balancing between correctness and sparsity/closeness of the final counterfactual (i.e., allowing "non-optimal" explanations). Furthermore, as mentioned in the previous response, most algorithms used in practice consider the WACHTER-CFE formulation, instead of the MCR formulation. We therefore believe that our study could serve as a starting point for others to move away from the "strict" MCR problem formulation, aligning more closely to the formulations commonly used in practice and by this hopefully getting a better understanding of when and why commonly used heuristics work well in practice. Finally, we believe that it might be interesting to identify data points or regions in data space for which solving the WACHTER-CFE problem becomes feasible, probably not only depending on the parameterized classifier/regressor, but also on $lambda$ specifying the degree of non-optimality that we allow. This means moving away from worst-case analyses, as you suggest.
> >
> > In the revised manuscript, we added (in Section 5) a discussion on this as a potential future research direction for the community.
> >
> > ## Requested changes
> >
> > >Explicitly Bridge the Classifier-Regressor Gap: The paper would be significantly strengthened if the authors, in Section 4, explicitly justify the shift to a regressor formulation for their new proofs. They should add a discussion on how the inapproximability of the WACHTER-CFE regressor problem translates to the "classic" CFE classifier problem (MCR, Eq. 3). For instance, can any MCR problem for a classifier be reduced to this regressor formulation, preserving the approximation hardness?
> >
> > Resolved together with the next requested change.
> >
> > >Clarify the Choice of WACHTER-CFE: A brief paragraph could be added to Section 4 explaining why the WACHTER-CFE formulation (Eq. 2) was chosen for the inapproximability proofs over the MCR formulation (Eq. 3). This would help integrate the new contribution with the surveyed literature more tightly.
> >
> > In the revised manuscript (Section 4), we added a clarifying statement about this and the previous (i.e., first) requested change to the paragraph right after Definition 2.
> >
> > >Refine the Discussion on Practicality: The discussion in Section 5 could be expanded to be more actionable. Instead of just stating that heuristics are used, the authors could suggest that future work should move beyond worst-case complexity. They could propose analyzing the average-case complexity or parameterised complexity (e.g., complexity in terms of model depth rather than just total size) of these problems, which might better explain why heuristics are often "good enough" in practice.
> >
> > In the revised manuscript (in Section 5), we added a new paragraph on this, and also a very brief statement in the conclusion in Section 6.
> >
> > >Contextualise "Plausibility" and "Robustness": The paper rightly identifies that the complexity of "plausible" and "robust" explanations is a key open area. It would be beneficial to add a sentence acknowledging that the definitions used in the literature so far (e.g., "learning a separate function") are just a few of many possible formalisations. This would frame the "missing entries in Tables 1 and 2" as an even richer field for future work, covering other definitions like causal plausibility or manifold-based robustness.
> >
> > In the revised manuscript, we added a short statement acknowledging the existence and relevance of other formulations to the end of Section 2.1. Furthermore, we also added a brief acknowledging statement about this to the conclusion in Section 6, when referring to the "missing entries in Tables 1 and 2", highlighting the richness of the field for future work.

---

> ### Comment · Action_Editor_nC6z · 2025-11-15
> **Please engage with the author response!**
>
> Hi Reviewer RLgA,
>
> The authors have carefully responded to your review and have also provided a revision of their paper. I cannot accept any recommendation you make on the paper without seeing your good faith effort to engage with them in discussion.
>
> Best,
> Taylor

---

> > ### Comment · Reviewer_RLgA · 2025-11-15
> > **I will get to this by next Friday**
> >
> > Dear All,
> >
> > I am overwhelmed with the current rebuttals and submission cycles. Will respond to the authors constructively by next week.
> >
> > Kind Regards,
> > Reviewer RLgA

---

> > ### Comment · Reviewer_RLgA · 2025-11-20
> >
> > Dear Taylor,
> >
> > I have gone through the author responses, and they have clarified all my questions and made the requested changes in the revised version of the paper. Some of my questions were better resolved through the rebuttals of the other reviewers, and I am satisfied with the changes. I am leaning towards accept as this is an important problem being addressed, with more practical applications backing the theory of semi-counterfactuals to follow this up.
> >
> > Kind Regards,
> > Reviewer RLgA

---

> > > ### Comment · Action_Editor_nC6z · 2025-11-20
> > > **Great!**
> > >
> > > I was going to follow up here and recommend that you address these comments to the authors but it appears that you've made them public already! Great!
> > >
> > > Thank you for your efforts on this paper's behalf!

---

### Comment · Action_Editor_nC6z · 2025-11-03
**Let's continue with a great discussion phase for this paper!**

Hi everyone,

First off, I want to thank the authors for their patience while we worked hard to get a third review in place for this paper. This took a far longer time than I've ever experienced before with TMLR.

With the authors providing detailed responses to the reviews, as well as a clear revision of the paper in response to the feedback received, I want to encourage all of us to be as engaged as possible in a productive discussion phase. This is a great time to review the revised paper to see if the author's updates address the concerns you raised previously. Please continue the discussion on points of possible misunderstanding that the authors have clarified in their responses to you as well.

I feel that we owe it to each other to engage productively to determine whether this paper is ready for publication now or whether there are further revisions it may need. In the latter case, we can do our best to highlight what can be amended for such a resubmission.

Thanks!
-Taylor

---

### Decision · Action_Editor_nC6z · 2025-12-15

**Recommendation:** Accept as is

**Additional Comments:**

Thank you for your patience during the review of this work. It took far longer than we anticipated due to several instances of needing to find a new reviewer. This work has deserved full acceptance for several months but these delays have made it difficult. I apologize on behalf of TMLR. Thanks again for being great supports throughout the process.

**Audience:**

Yes

**Audience Explanation:**

The problem of counterfactual and semi-factual estimation is of significant importance given the rise of large language models and data privacy concerning these models, and there exists a lack of interpretability surrounding these models. This paper makes an attempt to solve that, by grouping together all the existing models and stating their respective complexities (Tables 1 and 2). An impactful contribution.

 - The study of the computational complexity of explainability methods in AI (and xAI in general) is a highly relevant topic with current impact.
- This paper addresses not only counterfactual explanations but also semi-factual ones (a less explored area).

**Claims And Evidence:**

Yes

**Claims Explanation:**

This paper examines the computational complexity of generating counterfactual and semi-factual explanations. It provides a concise introduction to both types of explanations and fundamental notions in computational complexity, following with a survey of known results on the computational hardness of counterfactual generation and semi-factual generations across various model classes.

The authors provide experiments to demonstrate the computational limitations of prior methods while also pointing toward a new inapproximability result about the explanations models can generate in counterfactual and semi-factual settings.